



# Estimating the potential cooling effect of cirrus thinning achieved via the seeding approach

Jiaojiao Liu[1] and Xiangjun Shi[1]

[1]School of Atmospheric Sciences, Nanjing University of Information Science and Technology, 210044, Nanjing, China

*Correspondence to*: Xiangjun Shi (shixj@nuist.edu.cn)

**Abstract.** Cirrus thinning is a newly emerging geoengineering approach to mitigate global warming. To sufficiently exploit the potential cooling effect of cirrus thinning with the seeding approach, a flexible seeding method is used to calculate the optimal seeding number concentration, which is just enough to prevent homogeneous ice nucleation from occurring. A simulation using the Community Atmosphere Model version 5 (CAM5) with the flexible seeding method shows a global

cooling effect of $1.36 \pm 0.18$ W m$^{-2}$, which is approximately two-thirds of that from artificially turning off homogeneous nucleation ($-1.98 \pm 0.26$ W m$^{-2}$). However, simulations with fixed seeding ice nuclei particle number concentrations of 20 and 200 L$^{-1}$ show a weak cooling effect of $-0.27 \pm 0.26$ W m$^{-2}$ and warming effect of $0.35 \pm 0.28$ W m$^{-2}$, respectively. Further analysis shows that cirrus seeding leads to a significant warming effect of liquid and mixed-phase clouds, which counteracts the cooling effect of cirrus clouds. This counteraction is more prominent at low latitudes and leads to a pronounced net warm

effect over some low latitude regions. The sensitivity experiment shows that cirrus seeding carried out at latitudes with solar noon zenith angles greater than 12° could yields a stronger global cooling effect of $-2.00 \pm 0.25$ W m$^{-2}$. Overall, the potential cooling effect of cirrus thinning is considerable, and the flexible seeding method is essential.

## 1 Introduction

Global warming has been proven by observations and has demonstrated many adverse effects on the environment and economy

(Alexander et al., 2006; Feely et al., 2009; Lenton et al., 2019; Milne et al., 2009; Myhre et al., 2013). Conserving energy and reducing greenhouse gas emissions are regarded as the primary strategies to counteract global warming, but these strategies may not be satisfactory (Fuss et al., 2018; IEA, 2019; Rogelj et al., 2015; Solomon et al., 2009). Therefore, geoengineering as a back-up tool to against climate warming has been receiving increasing attention in recent years (e.g., Gasparini et al., 2020; Jones et al., 2018; Keith and MacMartin, 2015; Lawrence et al., 2018; Lohmann and Gasparini, 2017; Macnaghten and Owen,

2011). Geoengineering is usually divided into two categories: carbon dioxide removal (CDR), which aims to permanently eliminate $CO_2$ from the atmosphere, and solar radiation management (SRM), which proposes artificial intervention in the radiation budget (Caldeira et al., 2013; Heutel et al., 2018; Irvine et al., 2016; Kravitz et al., 2011; Vaughan and Lenton, 2011). It is well known that cirrus clouds (ice clouds) typically reflect less incoming solar radiation but block more of Earth's outgoing longwave radiation, which warm our planet (Berry and Mace, 2014; Hong et al., 2016; Matus and L'Ecuyer, 2017). Cirrus



thinning geoengineering, which allows more longwave radiation to escape into space so that cool the Earth, is investigated as
a new SRM approach and has been proposed in Geoengineering Model Intercomparison Project Phase 6 (GeoMIP6, Kravitz
et al., 2015). In GeoMIP6, cirrus thinning is simulated by artificially increasing the sedimentation velocity of ice crystals (ICs).
Simulations with this idealized approach indicate that cirrus thinning can produce the desired globally averaged cooling effect
(~ −2.0 W m$^{-2}$, e.g., Gasparini et al., 2017; Jackson et al., 2016; Muri et al., 2014). Considering the physical feasibility,

simulating cirrus thinning by seeding with ice nuclei particles (INPs) is a better approach that can prevent homogeneous
nucleation from occurring, thereby decreasing the number concentration of ICs (e.g., Gruber et al., 2019; Storelvmo and Herger,
2014; Storelvmo et al., 2013). Previous studies have shown that the cooling effect achieved via the seeding approach is
sensitive to the seeding number concentration ($N_{seed}$), and even the strongest cooling effect may not be ideal (above −1.0 W
m$^{-2}$, e.g., Gasparini and Lohmann, 2016; Gasparini et al., 2017; Penner et al., 2015). Note that $N_{seed}$ used in these model

simulations is fixed (usually in the range of 0.1 to 200 L$^{-1}$), and the seeding strategy is uninterrupted (i.e., seeding occurs at
every model time step). This study shows that the potential cooling effect of cirrus thinning cannot be sufficiently exploited
due to the fixed seeding method. Moreover, a flexible seeding method is introduced to calculate the optimal $N_{seed}$ ($N_{seedopt}$)
based on the cirrus formation condition. The major purpose of this study is to estimate the potential cooling effect of cirrus
thinning achieved via the seeding approach.

In this study, the cooling effects of cirrus thinning with different seeding methods are estimated. The paper is organized as
follows. The flexible seeding method and its advantages are introduced in Sect. 2. This section also introduces the models and
experimental designs that are employed. Comparisons of the cooling effects among different seeding methods and the main
mechanism for the cooling effect are presented in Sect. 3. Finally, Sect. 4 provides the conclusions and discussions.

## 2 Methods and experiments

### 2.1 Cirrus thinning by seeding with ice nuclei particles

To better understand the seeding methods used in this study, it is necessary to briefly introduce the mechanism of cirrus
thinning by seeding with INPs. In cirrus clouds, ICs are formed by homogeneous nucleation on soluble aerosol particles or
heterogeneous nucleation on insoluble aerosol particles (Pruppacher and Klett, 1998). As ice-phase supersaturation ($S_i$) rises,
heterogeneous nucleation occurs earlier with the aid of INPs (i.e., insoluble aerosols). A few ICs (usually less than 100 L$^{-1}$)

are generated due to the relatively low number concentration of INPs. These newly formed ICs consume water vapor via
deposition growth and then hinder $S_i$ from rising (DeMott et al., 2003; Hoose and Möhler, 2012; Lohmann et al., 2008). The
threshold $S_i$ for homogeneous nucleation ($S_{ihom}$) is relatively higher. Therefore, homogeneous nucleation cannot occur (i.e., $S_i$
cannot reach $S_{ihom}$) if there are enough newly formed ICs from heterogeneous nucleation. However, homogeneous nucleation
can produce a large number of ICs once it takes place (usually much greater than 100 L$^{-1}$) because the number concentration

of soluble aerosols in the upper troposphere is abundant (Barahona and Nenes, 2009; Kärcher, 2002). According to the
competition between homogeneous and heterogeneous nucleation, seeding with a few INPs (usually less than 100 L$^{-1}$) can



produce more ICs from heterogeneous nucleation and then inhibit homogeneous nucleation (Barahona and Nenes, 2009; Liu and Penner, 2005, McGraw et al., 2020). As a result, the in-cloud IC number concentrations ($N_i$) are usually decreased (i.e., cirrus thinning, Storelvmo and Herger, 2014; Storelvmo et al., 2013).

## 2.2 Models and parameterizations

In this study, we use a cloud parcel model to illustrate the impact of seeding on the ice nucleation process. The parcel model presents the ICs formation process in an adiabatically rising air parcel with a constant updraft vertical velocity ($W$). Equations that describe the evolution of temperature ($T$), pressure ($P$), ice water mixing ratio ($Q_i$), and ice particle size ($R_i$) can be found in Pruppacher and Klett (1998). The $S_i$ is diagnosed from the conservation equation of total water (i.e., water vapor plus ice water). The homogeneous nucleation rate ($J$) of sulfate aerosol particles is calculated based on the water activity (Koop et al., 2000). The heterogeneous frozen fraction of dust aerosol particles is calculated by the classical nucleation theory (CNT, following Barahona and Nenes, 2009) with a freezing efficiency of 1.0 (i.e., 100% dust aerosols can act as INPs). More details about this cloud parcel model can be found in Shi and Liu (2016).

The climate model used in this study is version 5.3 of the Community Atmosphere Model (CAM5; Neale, 2012) with an improved ice nucleation parameterization that considers the effect of pre-existing ICs and the in-cloud vertical velocity fluctuations (Shi et al., 2015; Shi and Liu, 2016). The ice nucleation parameterization considers the competition between heterogeneous freezing on coarse-mode dust aerosol particles and homogeneous freezing on sulfate aerosol particles. Here, 100% coarse-mode dust aerosols can act as INPs (Liu and Penner, 2005; Shi et al., 2015). Considering that sulfate aerosol particles may transform into glassy at very low temperatures (Murray et al., 2010), homogeneous nucleation is switched off below −68 ℃ (~ 205 K). Note that there is no homogeneous nucleation in mixed-phase clouds (0℃ ≥ $T$ > −37 ℃). The sub-grid vertical velocity ($W_{sub}$) derived from the turbulent kinetic energy is used to drive ice nucleation parameterization (Gettelman et al., 2010). The effect of pre-existing ICs on ice nucleation is parameterized by reducing the vertical velocity for ice nucleation ($W_{pre}$; Barahona et al., 2014; Kärcher et al., 2006; Shi et al., 2015). In the improved ice nucleation parameterization, the newly nucleated IC number concentration ($N_{inuc}$) is calculated as a function of $T$, $P$, $S_i$, $W_{sub}$, $W_{pre}$, the number concentration of coarse-mode dust aerosols ($N_{dust}$), and the number concentration of sulfate aerosols ($N_{sul}$). The cloud microphysics is represented by a two-moment scheme (Morrison and Gettelman, 2008).

## 2.3 Flexible seeding method

According to the mechanism of cirrus thinning caused by seeding with INPs, it is clear that $N_{seedopt}$ is the minimal number concentration, which is just enough to prevent homogeneous nucleation from occurring. If $N_{seed}$ is less than $N_{seedopt}$ (underseeding), the newly formed ICs from heterogeneous nucleation are insufficient; thus, homogeneous nucleation still occurs and produces a relatively large $N_{inuc}$. If $N_{seed}$ is larger than $N_{seedopt}$ (overseeding), despite homogeneous nucleation being suppressed, $N_{inuc}$ remains somewhat larger due to excessive $N_{seed}$. Notably, in terms of consuming water vapor and hindering homogeneous nucleation, it is clear that ICs are superior to INPs. In other words, ICs can act as cheaper, cleaner, and safer





INPs. Therefore, ICs are used as the seeding material in the flexible seeding method introduced by this study. The formulas for calculating $N_{seedopt}$ are introduced in the Appendix. The $N_{seedopt}$ is a function of cirrus ambient conditions, aerosol properties, and the radius of seeding ICs ($R_{seed}$). The $R_{seed}$ is a tunable given parameter. Notably, seeding with ICs occurs only where homogeneous nucleation would occur without seeding (i.e., flexible seeding strategy).

The left panel in Fig. 1 illustrates the advantage of $N_{seedopt}$. Parcel model results show that without seeding (REF, black lines), heterogeneous nucleation takes place at $S_i$ >10% and produces 10 L$^{-1}$ of ICs. Because these newly formed ICs are too few to prevent $S_i$ from increasing, homogeneous nucleation takes place at $S_i > S_{ihom}$ (~ 56%) and produces a large number of ICs (2937 L$^{-1}$). The final $N_i$ (i.e., $N_{inuc}$) is 2947 L$^{-1}$. In the simulation with pure heterogeneous nucleation (HET, green lines), the final $N_i$ is 10 L$^{-1}$. In the simulation that seeding with 28 L$^{-1}$ ($N_{seedopt}$ is 28 L$^{-1}$) of ICs (OPT, red lines), the newly formed ICs from heterogeneous nucleation (10 L$^{-1}$) and seeding ICs are just enough to prevent $S_i$ from reaching $S_{ihom}$. The final $N_i$ (i.e., $N_{inuc}$ + $N_{seedopt}$) is 38 L$^{-1}$. In the simulation that seeding with 20 L$^{-1}$ of coarse-mode dust aerosol particles (INP20, blue lines, underseeding), heterogeneous nucleation produces more ICs (30 L$^{-1}$) than the REF simulation. However, homogeneous nucleation still occurs and produces 715 L$^{-1}$ of ICs. The final $N_i$ is 745 L$^{-1}$. In the simulation that seeding with 200 L$^{-1}$ of coarse-mode dust aerosol particles (INP200, orange lines, overseeding), the newly formed ICs from heterogeneous nucleation (210 L$^{-1}$) are large enough to prevent homogeneous nucleation from occurring. The final $N_i$ is 210 L$^{-1}$. Overall, seeding with INPs/ICs can lead to a lower $N_i$, and $N_i$ from the OPT simulation is closest to the HET simulation.

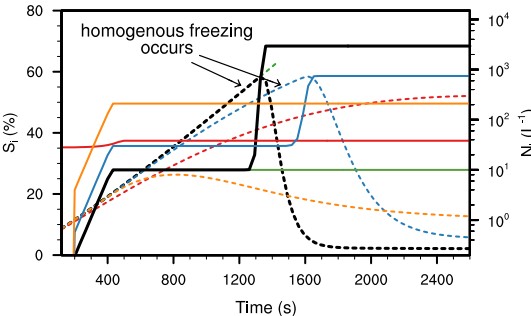

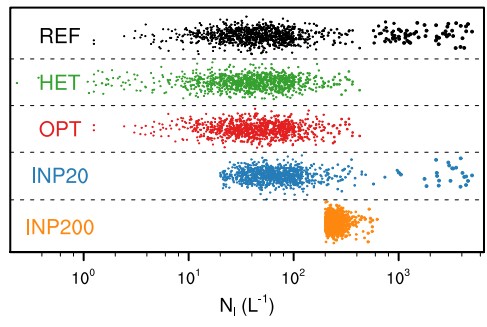

**Figure 1: Schematic diagram of different seeding methods, reference results without seeding (REF, black), pure heterogeneous nucleation (HET, green), seeding with optimal number concentration of ICs (OPT, red), seeding with 20 L$^{-1}$ of INPs (INP20, blue), and seeding with 200 L$^{-1}$ of INPs (INP200, orange). The optimal seeding method uses ICs with the radius of 25 µm. The left panel shows simulation results from the parcel model with given initial conditions ($P$=330 hPa, $T$=220 K, $W$=0.3 m s$^{-1}$, $N_{dust}$=10 L$^{-1}$, and $N_{sul}$=500 000 L$^{-1}$). The solid lines denote the total number concentrations of ICs in the parcel ($N_i$, units: L$^{-1}$), which include the seeding ICs and the newly formed ICs, and the dashed lines denote ice supersaturation ($S_i$, units: %). The arrows point to the beginning of homogeneous nucleation. The right panel shows the $N_i$ from ice nucleation parameterizations driven by the same 1000 datasets of input variables (one dot denotes one offline result), which are sampled from the CAM5 simulation. The horizontal coordinate axis is $N_i$, and the vertical coordinate axis is meaningless.**

Additionally, we run large-ensemble ice nucleation offline experiments to show advantage of the flexible seeding strategy (Fig. 1, right panel). A total of 1000 cirrus formation cases are sampled from the CAM5 simulation without seeding. The input variables ($T$, $P$, $S_i$, $W_{sub}$, $N_{dust}$, and $N_{sul}$) used to drive ice nucleation parameterization in CAM5, are used to drive these offline experiments. The homogeneous nucleation events account for 7.4 % (i.e., 74 homogeneous nucleation cases). Five experiments





corresponding to the parcel model simulations are carried out. Each experiment is driven by the same 1000 cases. In the two

fixed seeding experiments (i.e., the INP20 and INP200 experiments), the INPs (i.e., coarse-mode dust aerosols) are added for all 1000 cases even if there were no homogeneous nucleation (i.e., uninterrupted seeding strategy). Compared with the REF experiment, all large $N_i$ cases (dots with $N_i > 500$ L$^{-1}$) totally vanish in the HET experiment because only heterogeneous nucleation events occur. The $N_i$ distribution in the OPT experiment is similar to that in the HET experiment except for some low $N_i$ (<10 L$^{-1}$) cases. In the INP20 experiment, there are some large $N_i$ cases because homogeneous nucleation still occurs

in 36 cases. In the INP200 experiment, there are no large $N_i$ cases because almost all homogeneous nucleation cases are suppressed, whereas the $N_i$ from all cases is greater than 200 L$^{-1}$ due to the large $N_{seed}$. In short, the flexible seeding method is better than the fixed seeding method.

**2.4 Experimental setups**

The CAM5 model experiments are carried out to estimate the cooling effect of cirrus thinning. Table 1 summarizes all the

experiments performed in this study. The REF, HET, OPT, INP20, and INP200 experiments corresponding to the offline experiments discussed above. In the INP20 and INP200 experiments, $N_{dust}$ used for driving ice nucleation parameterization (cirrus clouds only) increase by 20 and 200 L$^{-1}$, respectively. Note that $N_{dust}$ in the aerosol module is not changed. In other words, the seeding INPs only impact the ice nucleation process. In the OPT experiment, the seeding ICs are directly added into the cloud microphysics scheme. As a result, these seeding ICs would affect both the ice nucleation process and other cloud

microphysics processes.

**Table 1. List of CAM5 experiments**

| Experiments | Description |
|---|---|
| REF | Reference experiment without cirrus thinning. |
| Cirrus thinning with different methods | |
| HET | Pure heterogeneous nucleation, homogeneous nucleation is artificially turned off. |
| OPT | Implement seeding globally using the flexible seeding method with ICs radius ($R_{seed}$) of 50 μm. |
| INP20 | Implement seeding globally with 20 L$^{-1}$ of INPs. |
| INP200 | Implement seeding globally with 200 L$^{-1}$ of INPs. |
| Sensitivity experiments for the flexible seeding method | |
| R10 | Similar to OPT but $R_{seed}$ is set to 10 μm. |
| GT | Similar to OPT but seeding occurs over target regions, where the solar noon zenith angles are greater than 12°. |

Additionally, we set up two sensitivity experiments for the flexible seeding method (Table 1). First, the tunable parameter $R_{seed}$ is investigated. The $R_{seed}$ is 10 μm in the R10 experiment, whereas the $R_{seed}$ is 50 μm in the OPT experiment. Second, the





seeding region is investigated. Cirrus thinning also leads to more incoming solar radiation (warming effect), which counteracts the cooling effect from more outgoing longwave radiation, especially for the low solar noon zenith angle regions (Storelvmo and Herger, 2014; Storelvmo et al., 2014). Furthermore, this study also finds that the cooling effect over low latitude regions is less susceptible to cirrus seeding for other reasons (see Sect. 3.1). Thus, another sensitivity experiment with a specific geographical target (i.e., the GT experiment) is examined. Similar to the study of Storelvmo and Herger (2014), cirrus seeding

is only carried out at latitudes where the solar noon zenith angles are greater than 12°, which are approximately 80% of the Earth's surface.

In this study, all CAM5 experiments are atmosphere-only simulations with the same prescribed climatological ocean surface conditions. All experiments run for 11 model years at a horizontal resolution of 1.9°×2.5° and model time step of 30 min. The first year is considered to be a spin-up period, and the last 10 years are used in the analyses. The standard deviations, which

are estimated from the averages of each year, are used for variability analysis.

## 3 Estimating the cooling effect of cirrus thinning

### 3.1 Comparisons among different seeding methods

First, we analyze the impact of cirrus seeding on the ice nucleation process (Fig. 2). The contribution of homogeneous nucleation to cirrus formation ($F_{hom}$) is essential for the radiative properties of cirrus clouds (Jensen et al., 2013; Shi and Liu,

2016). Here, $F_{hom}$ is quantified as the ratio of the homogeneous nucleation occurrence frequency to the ice nucleation occurrence frequency ($F_{nuc}$). In the REF experiment, $F_{hom}$ is low near dust source regions (e.g., the Saharan Desert and Arabian Desert). The $F_{hom}$ is high over other tropical regions due to the large $W_{sub}$ (not shown). Generally, $F_{hom}$ is low (< 20%) in most regions, which is consistent with observations that heterogeneous nucleation is the dominant mechanism for cirrus formation (Cziczo et al., 2013; Jensen et al., 2013). Although $F_{hom}$ from the INP20 experiment is decreased substantially, there are still

some homogeneous freezing events (3.38% of all cirrus and 5.20% at 233 hPa). In the INP200 experiment, there are only a few homogeneous freezing events (0.42% of all cirrus and 0.63% at 233 hPa) due to the larger $N_{seed}$ of INPs. Both the INP20 and INP200 experiments show that the averaged number concentration of ICs produced from heterogeneous freezing events ($N_{ihet}$) is increased. This increase would lead to more intense competition between homogeneous and heterogeneous nucleation. As a result, the averaged number concentration of ICs produced from homogeneous freezing events ($N_{ihom}$) from the INP20

and INP200 experiments are substantially decreased compared with the REF experiment. As expected, $F_{hom}$ and $N_{ihom}$ are zero from the HET and OPT experiments. It is noteworthy that a large number of small ICs (e.g., homogeneous nucleation occurs) would exist for a long time, consuming water vapor via deposition growth and then hindering the subsequent ice nucleation (Shi et al., 2015). Therefore, $F_{nuc}$ from the REF experiment is very low (< 4%) in most regions, and $F_{nuc}$ from the cirrus thinning experiments (i.e., the HET, OPT, INP20, and INP200 experiments) are obviously increased due to the decreases in $F_{hom}$ and

$N_{ihom}$. This finding suggests that the impact of cirrus seeding (including the HET experiment) on the ice nucleation process is very complicated. There is not only the direct instantaneous impact but also the indirect impact caused by subsequent changes.



**Figure 2:** Annual zonal mean and 233 hPa spatial distribution of the homogeneous nucleation contribution to cirrus formation ($F_{hom}$, first panel), averaged ICs number concentration produced from heterogeneous freezing events ($N_{ihet}$, second panel) and from homogeneous freezing events ($N_{ihom}$, third panel), and ice nucleation occurrence frequency ($F_{nuc}$, last panel). Experimental names are shown in the upper left corner, and globally mean values are shown in the upper right corner. The two black lines are 0 and −37 °C isotherms. The results are sampled from model grids where $F_{nuc}$ are greater than 0.1 %.





Figure 3 shows the impact of cirrus seeding on cloud properties. In the cloud microphysics scheme, the in-cloud IC number concentration (i.e., $N_i$) mainly depends on the ice nucleation process (i.e., $N_{inuc}$, Shi et al., 2015; Shi and Liu, 2016). Therefore,

the annual averaged $N_i$ from the cirrus thinning experiments are decreased significantly in most cirrus clouds (ice clouds), especially from the HET and OPT experiments. However, $N_i$ from the HET and OPT experiments are increased in the lower mixed-phase clouds. The reason might be that the averaged sizes of cirrus ICs from the HET and OPT experiments are increased in the upper troposphere (not shown), and it becomes easier for these larger ICs to fall into mixed-phase clouds. The $N_i$ from the INP20 experiment is not significantly decreased over the tropical regions because there are still many homogeneous

freezing events ($F_{hom}$ and $N_{ihom}$ in Fig. 2). Compared with the REF experiment, $N_i$ from the INP200 experiment is obviously increased in the tropical upper troposphere and the polar troposphere. This increase is because the homogeneous nucleation contribution (i.e., $F_{nuc} \times F_{hom} \times N_{ihom}$) from the REF experiment is relatively low, and the heterogeneous nucleation contribution (i.e., $F_{nuc} \times N_{ihet}$) from the INP200 experiment increases dozens of times over these regions (Fig. 2). Similarly, the vertically integrated $N_i$ (i.e., column $N_i$) from the HET and OPT experiments are significantly decreased in most regions. In contrast, the

changes in column $N_i$ from the INP20 and INP200 experiments are not notable. The changes in ice water content (IWC) and ice water path (IWP) from the INP20 and INP200 experiments are also non-significant in most regions. In the HET and OPT experiments, the relative increases in IWC in mixed-phase clouds are stronger than the relative increases in $N_i$ because the averaged radii of mixed-phase cloud ICs are increased (not shown). Although the increases in IWC in mixed-phase clouds counteract the decreases in IWC in ice clouds to some extent, the IWP are still significantly decreased in most regions from

the HET and OPT experiments. However, the IWP are significantly increased over a few regions (e.g., middle Africa and northern Brazil) because the decreases in IWC in ice clouds are slight and even smaller than the increases in IWC in mixed-phase clouds over there (not shown). The changes in liquid water content (LWC) and liquid water path (LWP) from the INP20 and INP200 experiments are non-significant in most regions, whereas both the LWC and LWP from the HET and OPT experiments are significantly decreased in some low- and mid-latitude regions. One possible reason is the falling ICs accrete

by riming of cloud droplets (Gasparini et al., 2017; Storelvmo et al., 2013), and the conversion efficiency of cloud droplets to precipitation is increased. Another possible reason is that cirrus thinning reduces atmospheric stability via the radiative budget, leading to stronger convective precipitation (Kristjánsson et al., 2015; Storelvmo and Herger, 2014; Storelvmo et al., 2013), which would consume more cloud water. The above analyses are in agreement with previous studies, which show that cirrus thinning might result in complex impacts on mixed-phase and liquid clouds (Gasparini and Lohmann, 2016; Gruber et al.,

210   2019).



**Figure 3:** Annual zonal mean of in-cloud IC number concentration ($N_i$, first row), ice water content (IWC, third row), and liquid water content (LWC, fifth row) from the REF experiment (first column) and the relative changes from the HET, OPT, INP20 and INP200 experiments with respect to the REF experiment (second to fifth columns). The corresponding spatial distributions of vertically integrated $N_i$ (Column $N_i$, second row), ice water path (IWP, fourth row), and liquid water path (LWP, sixth row) from the REF experiment and the differences ("Δ") from the HET, OPT, INP20 and INP200 experiments with respect to the REF experiment. Globally mean values are shown in the upper right corner, and the standard deviations calculated from the difference of each year for 10 years are shown in brackets. The shadow denotes that the differences between two experiments are not significant at the 95% level based on Student's t-test.



The cooling effect of cirrus thinning is usually quantified by the anomaly in cloud radiative effect (ΔCRE, Mitchell and Finnegan, 2009; Storelvmo and Herger, 2014). For the convenience of expression, "Δ" indicates the difference between the cirrus thinning experiments and the REF experiment. In addition to the model standard diagnostics of CRE, CRE from ice clouds (iCRE), mixed-phase clouds (mCRE), and liquid clouds (lCRE) are also diagnosed separately. Note that cirrus clouds are clouds at temperatures below –37 °C and above 440 hPa (Boucher et al., 2013), so we refer to them as ice clouds in this

study.

The iCRE and its shortwave ($iCRE_{SW}$) and longwave ($iCRE_{LW}$) components are analyzed first (Fig. 4). The globally averaged iCRE from the REF experiment is 6.49 W m$^{-2}$ (net warming effect) with a shortwave component ($iCRE_{SW}$) of −5.30 W m$^{-2}$ (cooling effect) and a longwave component ($iCRE_{LW}$) of 11.79 W m$^{-2}$ (stronger warming effect). This globally averaged iCRE is in the possible range compared with recent studies (4.5 ~ 6.8 W m$^{-2}$, Gasparini and Lohmann, 2016; Gasparini et al., 2020;

Hong et al., 2016; Lohmann and Gasparini, 2017; Muench and Lohmann, 2020). The globally averaged $iCRE_{SW}$ from the HET, OPT, INP20, and INP200 experiments increase (less negative, warming effect) by 3.39, 3.25, 0.38, and 0.30 W m$^{-2}$, respectively. The decrease in $iCRE_{LW}$ (cooling effect) from all cirrus thinning experiments are stronger, especially from the HET (−6.84 W m$^{-2}$) and OPT (−6.29 W m$^{-2}$) experiments. Although $\Delta iCRE_{LW}$ from the HET and OPT experiments show significant cooling effect over most regions, there are still a few regions with warming effects (middle Africa and northern

Brazil) due to higher ice cloud occurrence frequencies (not shown). The spatial patterns of $\Delta iCRE_{SW}$ and $\Delta iCRE_{LW}$ are generally in agreement with the changes in IWP and column $N_i$ (Fig. 3). In terms of ΔiCRE, the HET (−3.45 W m$^{-2}$) and OPT (−3.04 W m$^{-2}$) experiments show much stronger cooling effects than the INP20 (−0.44 W m$^{-2}$) and INP200 (−0.01 W m$^{-2}$) experiments. Following Gasparini et al. (2020), a diagnosed variable, the so-called cirrus seeding effectiveness, is used to show how many proportions of iCRE are eliminated by cirrus seeding (i.e., the absolute value of ΔiCRE divided by iCRE). The

globally averaged cirrus seeding effectiveness from the HET and OPT experiments are 56.19 % and 49.40 %, respectively. These values are much higher than those from the INP20 (6.69 %) and INP200 (−2.22 %) experiments. The fixed seeding method restricts the cirrus seeding effectiveness. Notably, over some tropical regions, the cirrus seeding effectiveness from the HET and OPT experiments are somewhat low, although the ΔiCRE are relatively strong (< −5 W m$^{-2}$). One reason is that iCRE is relatively strong (> 10 W m$^{-2}$), but convective detrainment (anvil cirrus, which is not influenced by cirrus seeding)

contributes more to iCRE (not shown). Another reason is that the ratio of $\Delta iCRE_{SW}$ to $\Delta iCRE_{LW}$ is higher over tropical areas due to the small solar noon zenith angles (not shown).

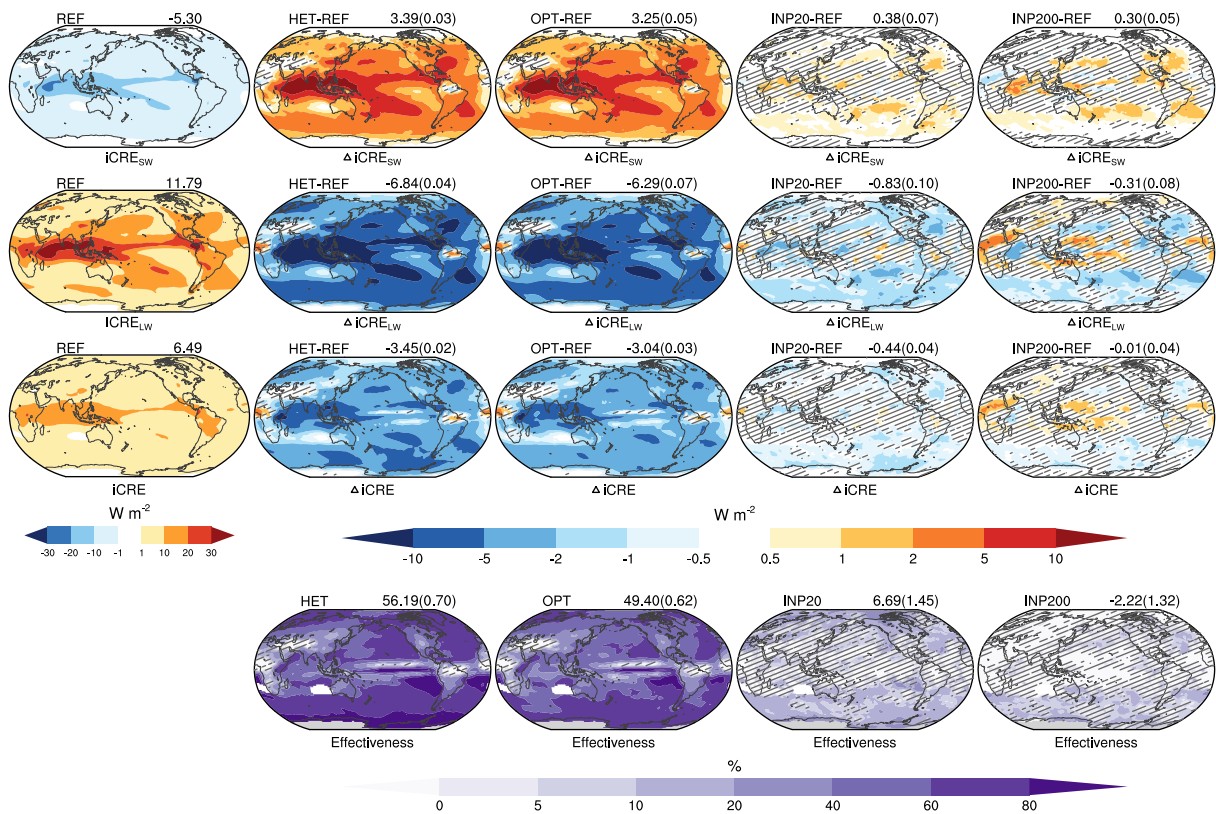

**Figure 4: The annual mean spatial distribution of ice cloud shortwave radiative effect (iCRE$_{SW}$, first row), ice cloud longwave radiative effect (iCRE$_{LW}$, second row), ice cloud radiative effect (iCRE= iCRE$_{SW}$+ iCRE$_{LW}$, third row), and cirrus seeding effectiveness (fourth row) from the REF experiment (first column) and the differences ("Δ") from the HET, OPT, INP20 and INP200 experiments (second to fifth columns). Note that regions with absolute value of iCRE that is < 1.0 W m$^{-2}$ from the REF experiment are excluded for calculating cirrus seeding effectiveness. Global mean values are shown in the upper right corner, and the corresponding standard deviations calculated from the difference of each year for 10 years are shown in brackets. The shadow denotes that the differences between two experiments are not significant at the 95% level based on Student's t-test.**

In addition to iCRE, mCRE and lCRE are also obviously influenced by cirrus thinning (Fig. 5). Compared with the REF experiment, mCRE from the HET and OPT experiments are significantly increased in most ocean regions. The corresponding globally averaged ΔmCRE are 1.06 and 1.09 W m$^{-2}$, respectively. This warming effect (i.e., positive ΔmCRE) mainly comes from the increasing longwave component (not shown), which is consistent with the increase in IWC in mixed-phase clouds (Fig. 3). The globally averaged lCRE from the HET and OPT experiments increase (warming effect) by 1.06 and 0.94 W m$^{-2}$, respectively. The ΔlCRE is strong (> 2 W m$^{-2}$) over some low- and mid-latitude regions that couple with the decreases in LWP (Fig. 3). Both ΔmCRE and ΔlCRE from the HET and OPT experiments show that the globally averaged values are several times larger than the corresponding standard deviations (0.11~0.14). This finding indicates that cirrus thinning with the HET/OPT method leads to a significant globally averaged warming effect from mixed-phase clouds (ΔmCRE) and liquid clouds (ΔlCRE), although ΔmCRE and ΔlCRE are not statistically significant in most regions. Unlike the HET and OPT experiments, both ΔmCRE and ΔlCRE from the INP20 and INP200 experiments are weak and full of uncertainties. The overall




cooling effect of cirrus thinning (i.e., $\Delta$CRE) from the HET and OPT experiments are $-1.98 \pm 0.26$ W m$^{-2}$ and $-1.36 \pm 0.18$ W m$^{-2}$, respectively (Fig. 5). Compared with the cooling effect of ice clouds (i.e., $\Delta$iCRE, Fig. 4), these values drop by approximately half due to the warming effect exerted by mixed-phase and liquid clouds. The INP20 and INP200 experiments show a weak cooling effect ($-0.27 \pm 0.26$ W m$^{-2}$) and even a small warm effect ($0.35 \pm 0.28$ W m$^{-2}$), respectively. It is clear

that cirrus seeding with the flexible method could produce a notable globally cooling effect, which is much better than the fixed methods. Furthermore, the cooling effect with the flexible seeding method is significant over most mid- and high-latitude regions. Some low latitude regions show a pronounced warming effect because cirrus seeding leads to a stronger warming effect introduced by mixed-phase and liquid clouds (i.e., $\Delta$mCRE and $\Delta$lCRE). This finding suggests that cirrus seeding over low latitude regions might be redundant.

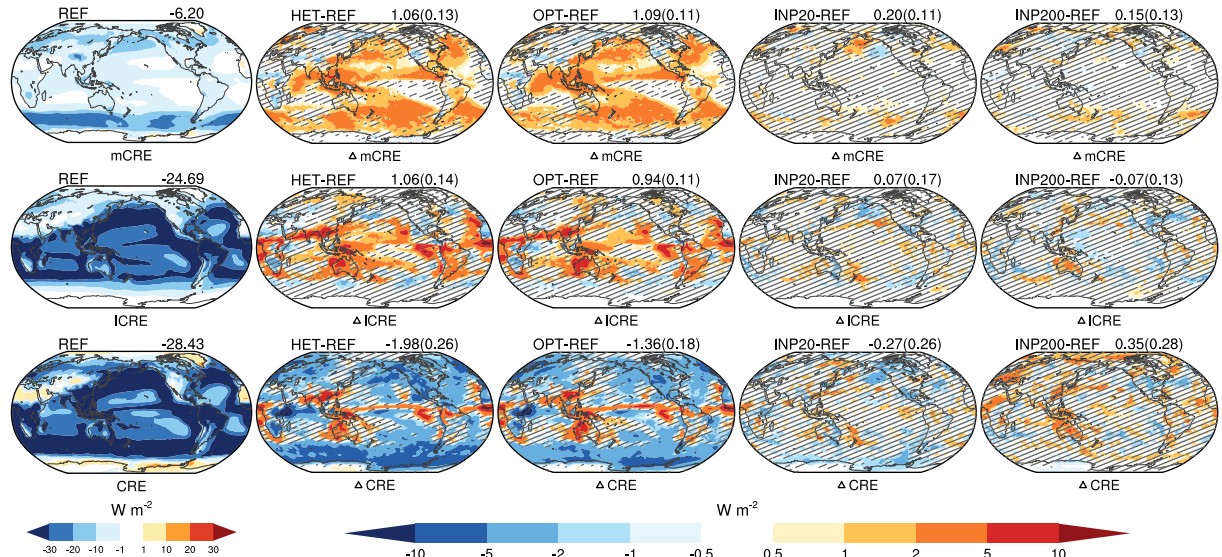


**Figure 5: Similar to Fig. 4 but for the mixed-phase cloud radiative effect (mCRE, first row), liquid cloud radiative effect (lCRE, second row), and all cloud radiative effect (CRE, third row).**

### 3.2 Sensitivity experiments regarding the flexible seeding method

To gain more understanding of cirrus thinning with the flexible seeding method, this section investigates the sensitivity

experiments of the cooling effect to $R_{seed}$ (R10 experiment) and the seeding region (GT experiment).

Figure 6 shows the seeding number concentration ($N_{seedopt}$) and seeding frequency ($F_{seed}$). As expected, the OPT and GT experiments show similar $N_{seedopt}$ in mid- and high-latitude regions. In these two experiments, $N_{seedopt}$ is less than 40 L$^{-1}$ in most regions. Because $N_{seedopt}$ increases with decreasing $R_{seed}$ (see Appendix), $N_{seedopt}$ from the R10 experiment is larger than that from the OPT and GT experiments. In these seeding experiments, it becomes easier for the ice nucleation process to reach

$S_{ihom}$ (i.e., cirrus seeding occurs) because the large amount of long-lived small ICs produced by homogeneous nucleation is cut off. As a result, $F_{seed}$ from the seeding experiments are much larger than the homogeneous freezing occurrence frequency ($F_{hom} \times F_{nuc}$) from the REF experiment (much less than 1%, Fig. 2). However, $F_{seed}$ from the seeding experiments is still relatively

low (< 4%) in most regions. The $F_{seed}$ from the GT experiment is even lower than 2% in most regions. The smaller ICs usually

have a longer lifetime in cirrus clouds, so $F_{seed}$ from the R10 experiment (1.01% of all cirrus and 1.05% at 233 hPa) is lower

than that from the OPT experiment (1.13% of all cirrus and 1.43% at 233hPa). Similar to the spatial distribution of $F_{hom}$ from

the REF experiment, $F_{seed}$ from the cirrus seeding experiments are much higher in the low latitude regions. This is the reason

why the globally averaged $F_{seed}$ from the GT experiment (0.82% of all cirrus and 0.80% at 233 hPa) is lower than that from

the OPT experiment.

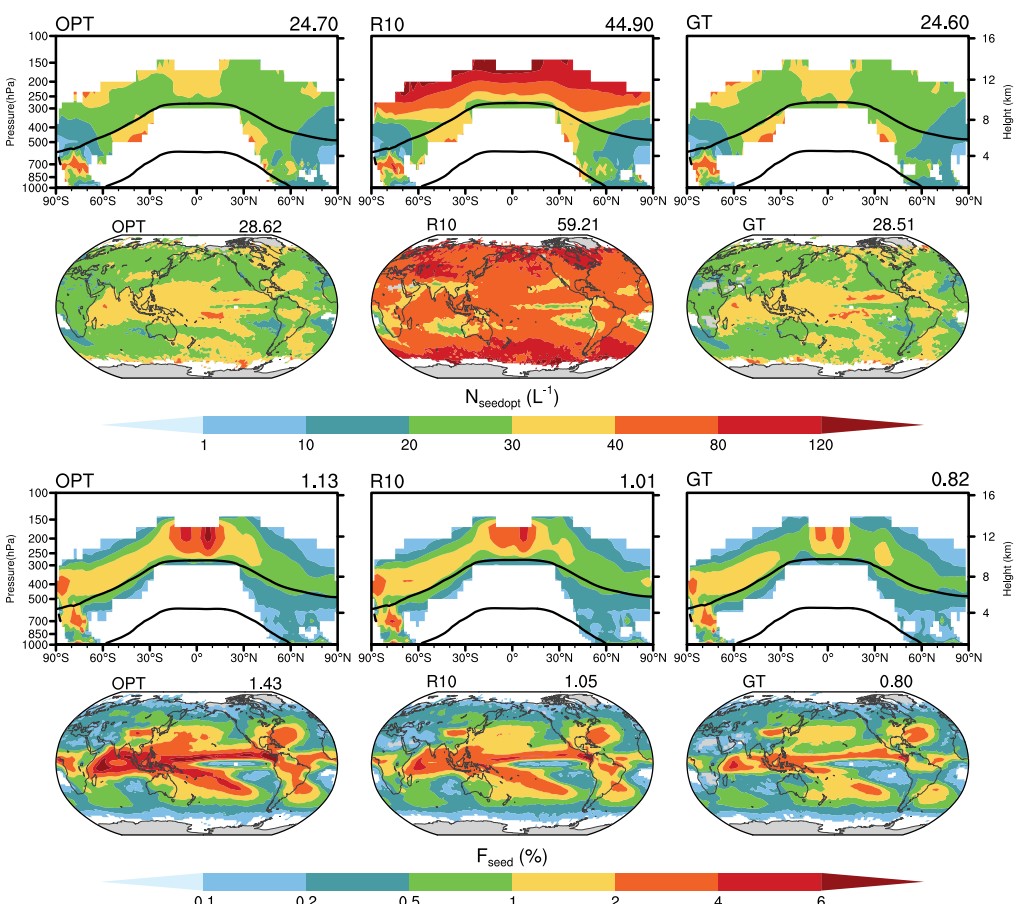

**Figure 6: Annual zonal mean and 233 hPa spatial distribution of the optimal seeding number concentration ($N_{seedopt}$, first panel) and seeding frequency ($F_{seed}$, second panel). The names of the experiments are shown in the upper left corner, and globally mean values are shown in the upper right corner. The results are sampled from model grids where $F_{seed}$ are greater than 0.1%.**

Figure 7 shows the cooling effects from the R10 and GT experiments. The globally averaged $\Delta iCRE$ from the R10 experiment

is $-2.58$ W m$^{-2}$. This ice cloud cooling effect is obviously weaker than that from the OPT experiment ($-3.04$ W m$^{-2}$) because

the seeding ICs in the R10 experiment (larger $N_{seedopt}$ and smaller $R_{seed}$) could live longer in cirrus clouds. Correspondingly,

the cirrus seeding effectiveness from the R10 experiment (43.02%) is also less than that from the OPT experiment (49.40%).

Similar to the OPT experiment, the R10 experiment also shows that cirrus seeding induces an obvious global warming effect



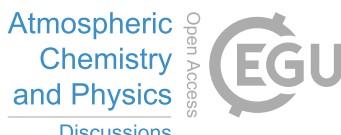

of mixed-phase and liquid clouds ($\Delta$mCRE and $\Delta$lCRE, not shown). Thus, the $\Delta$CRE from the R10 experiment is $-1.25 \pm 0.22$ W m$^{-2}$. This value is close to the OPT experiment ($-1.36 \pm 0.18$ W m$^{-2}$). In other words, the difference in the $\Delta$CRE (0.11 W

m$^{-2}$) between the OPT and R10 experiments is much less than the difference in $\Delta$iCRE (0.46 W m$^{-2}$). This finding indicates that the warming effects of mixed-phase and liquid clouds induced by seeding with smaller ICs become weaker. Compared with the OPT experiment, $\Delta$iCRE from the GT experiment becomes weaker over the regions without seeding (Figs. 7 and 4). Thus, the globally averaged $\Delta$iCRE only decreases by $-2.34$ W m$^{-2}$ from the GT experiment. Correspondingly, the cirrus seeding effectiveness from the GT experiment is also obviously less than that from the OPT experiment except for high-latitude

regions. As mentioned in Sect. 3.1, cirrus seeding would lead to a strong warming effect of mixed-phase and liquid clouds in low latitudes. As expected, in the GT experiment, this warming effect is constrained to some extent. The globally averaged cooling effect ($\Delta$CRE) from the GT experiment is $-2.00 \pm 0.25$ W m$^{-2}$, which is much stronger than that from the OPT experiment ($-1.36 \pm 0.18$ W m$^{-2}$) and even stronger than that from the HET experiment ($-1.98 \pm 0.26$ W m$^{-2}$). This finding suggests that cirrus seeding without low solar noon zenith angle regions might produce a better global cooling effect.

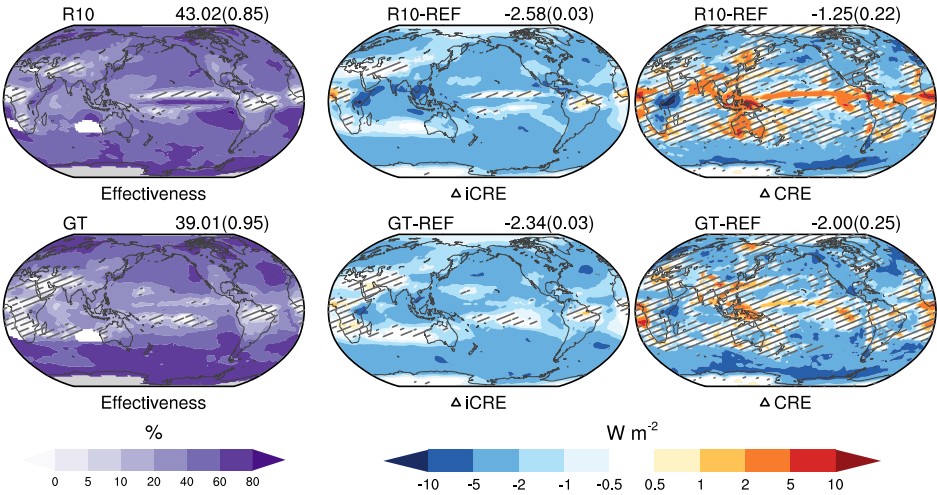

**Figure 7: Similar to Fig. 4 but for $\Delta$iCRE (middle), $\Delta$CRE (right), and cirrus seeding effectiveness (left) from the R10 (upper panel) and GT (lower panel) experiments.**

## 4 Conclusions and discussions

The major purpose of this study is to estimate the potential cooling effect of cirrus thinning. Based on the mechanism of cirrus

thinning by the seeding approach, a flexible seeding method is used to calculate the optimal seeding number concentration, which is just enough to prevent homogeneous ice nucleation from happening. Furthermore, the cirrus seeding approach could move further by injecting ice crystals (ICs) instead of ice nuclei particles (INPs). In terms of hindering homogeneous nucleation and environmental safety, ICs are better than INPs. More importantly, the problem of INPs transportation discussed in previous studies might be solved because ICs can be made from ambient atmospheric water vapor.





Both parcel model simulations and large-ensemble ice nucleation offline experiments show that the flexible seeding method
       has obvious advantages over the fixed seeding method. Furthermore, the CAM5 simulations with the flexible seeding method
       (implement seeding globally) show a notable global cooling effect, $-1.36 \pm 0.18$ W m$^{-2}$ from seeding with ICs of 50 μm (OPT
       experiment) and $-1.25 \pm 0.22$ W m$^{-2}$ from seeding with ICs of 10 μm (R10 experiment). However, simulations with fixed
       seeding number concentrations of 20 and 200 INPs L$^{-1}$ show weak cooling effects of $-0.27 \pm 0.26$ W m$^{-2}$ and warming effect

of $0.35 \pm 0.28$ W m$^{-2}$, respectively. Note that some previous work using CAM5 with the fixed seeding method showed notable
       cooling effect ($\sim -2$ W m$^{-2}$, e.g., Storelvmo and Herger, 2014; Storelvmo et al., 2014). This attributes to the contribution of
       homogeneous nucleation to cirrus formation ($F_{hom}$) from the default CAM5 model used in their study (Penner et al., 2015).
       The $F_{hom}$ from default CAM5 simulations is relatively higher because the default version neglects the effect of pre-existing
       ICs (Shi et al., 2015). Penner et al. (2015) tuned the main ice nucleation mechanism in CAM5 to limit $F_{hom}$ and found that

cirrus thinning with a fixed seeding number concentration cannot produce definite global cooling effect. In this study, $F_{hom}$ is
       also limited to a low level (Fig. 2). Our results with the fixed seeding method are similar to the study of Penner et al. (2015).
       However, with the benefits of the flexible seeding method, cirrus seeding could produce a considerable cooling effect.
       This study also analyses the main mechanism for the cooling effect achieved via cirrus seeding. Simulation results show that
       cirrus seeding not only impacts ice clouds but also significantly impacts mixed-phase and liquid clouds. In terms of ice clouds,

cirrus thinning with the flexible seeding method could lead to a notable cooling effect. However, cirrus seeding also leads to
       a significant warming effect of mixed-phase and liquid clouds, which counteracts the cooling effect of cirrus clouds. Because
       the counteraction is more prominent over low-latitude regions, the low-latitude regions are less susceptible to cirrus seeding.
       This finding agrees with the previous finding that cirrus thinning is more effective at mid and high latitudes because of more
       insolation caused by cirrus thinning when the sun is overhead (Storelvmo et al., 2014). The warming effect of liquid clouds

from the OPT experiment ($0.94 \pm 0.11$ W m$^{-2}$) is similar to the study of Gasparini et al. (2017, Table 5, $0.96 \pm 0.25$ W m$^{-2}$
       from the ECHAM-HAM model simulation that seeding with 1INP L$^{-1}$ of 50 μm). However, the warming effect of mixed-
       phase clouds from the OPT experiment ($1.09 \pm 0.11$ W m$^{-2}$) is several times stronger than that from their results ($0.15 \pm 0.10$
       W m$^{-2}$). The climatic response to cirrus seeding is complex and might differ among different climate models and seeding
       methods.

Sensitivity experiments regarding the flexible seeding method show that smaller seeding ICs leads to a weaker global cooling
       effect of ice clouds due to the larger seeding number concentration and smaller ICs. On the other hand, the warming effects of
       mixed-phase and liquid clouds are also reduced to some extent because it becomes difficult for smaller ICs to fall out of ice
       clouds. Thus, the global cool effect from seeding with smaller ICs ($-1.25 \pm 0.22$ W m$^{-2}$) is not obviously weaker than seeding
       with larger ICs ($-1.36 \pm 0.18$ W m$^{-2}$). Because there are larger warming effects of mixed-phase and liquid clouds over some

low-latitude regions, avoiding implementing seeding over there may obtain a better global cooling effect. Sensitivity
       experiment show that seeding carried out at latitudes with solar noon zenith angles greater than 12° yields a stronger global
       cooling effect of $-2.00 \pm 0.25$ W m$^{-2}$, which is close to that of artificially turning off homogeneous nucleation over the whole
       Earth ($-1.98 \pm 0.26$ W m$^{-2}$). In addition, we carried out sensitivity experiments with other threshold values (23.5°, 18°, and





8°). With increasing thresholds, the global cooling effect of ice clouds is decreasing, and the global warming effects of mixed-
phase and liquid clouds are also decreasing. The overall cooling effect with the threshold of 12° is best. In short, the global

cooling effect is more sensitive to seeding regions than to the radius of seeding ICs. It is still possible to enhance the global

cooling effect of cirrus thinning if seeding with more suitable regions and times. However, estimating the cooling effect of

cirrus seeding based on commercial airliners (i.e., the limited time and place) is more realistic. We plan to investigate this

method in the next step.


**Appendix: The formula of optimal seeding number concentration ($N_{\text{seedopt}}$)**

For the ice nucleation parameterization with the pre-existing IC effect, the seeding ICs are considered to be pre-existing ICs.

The optimal number concentration of ICs ($N_{\text{seedopt}}$) depends on the ice nucleation parameterization, especially for its treatment

of the pre-existing IC effect.

Without the pre-existing ICs or seeding ICs, the temporal evolution of $S_i$ is governed by (Kärcher et al., 2006):

$$\frac{dS_i}{dt} = a_1 S_i W - (a_2 + a_3 S_i)\frac{dQ_{\text{nuc}}}{dt}, \tag{A1}$$

where the parameters $a_1$, $a_2$, and $a_3$ only depend on the ambient temperature and pressure. $W$ is the updraft velocity, and $\frac{dQ_{\text{nuc}}}{dt}$

denotes the growth rate of newly nucleated ICs. To account for the effect of pre-existing ICs and seeding ICs, the deposition

growth of pre-existing ICs ($\frac{dQ_{\text{pre}}}{dt}$) and seeding ICs ($\frac{dQ_{\text{seed}}}{dt}$) are added in Eq. (A1):

$$\frac{dS_i}{dt} = a_1 S_i W - (a_2 + a_3 S_i)(\frac{dQ_{\text{nuc}}}{dt} + \frac{dQ_{\text{pre}}}{dt} + \frac{dQ_{\text{seed}}}{dt}), \tag{A2}$$

Equation (A2) can be rewritten as the following form:

$$\frac{dS_i}{dt} = a_1 S_i(W - W_{\text{pre}} - W_{\text{seed}}) - (a_2 + a_3 S_i)\frac{dQ_{\text{nuc}}}{dt}, \tag{A3}$$

$$W_{\text{pre}} = \frac{a_2 + a_3 S_i}{a_1 S_i}\frac{dQ_{\text{nuc}}}{dt}, \tag{A4}$$

$$W_{\text{seed}} = \frac{a_2 + a_3 S_i}{a_1 S_i}\frac{dQ_{\text{seed}}}{dt}, \tag{A5}$$

The effect of pre-existing ICs on ice nucleation can be taken as reducing the vertical velocity ($W_{\text{pre}}$, Barahona et al., 2014).

Details about how to calculate $W_{\text{pre}}$ are introduced in Shi et al. (2015). Here, the reduced vertical velocity from seeding ice

(i.e., $W_{\text{seed}}$) is similar to $W_{\text{pre}}$. The $W_{\text{seed}}$ is a function of seeding ice number concentration ($N_{\text{seed}}$) and its radius ($R_{\text{seed}}$). Assuming

all seeding ICs have the same $R_{\text{seed}}$, the growth rate is given by:

$$\frac{dQ_{\text{seed}}}{dt} = \frac{4\pi\rho_i}{m_w}N_{\text{seed}}\frac{b_1 R_{\text{seed}}^2}{1 + b_2 R_{\text{seed}}}, \tag{A6}$$

where $\rho_i$ is the ice density and $m_w$ is the mass of a water molecule. $b_1 = \alpha v_{\text{th}} n_{\text{sat}} (S_i - 1)/4$, $b_2 = \alpha v_{\text{th}} n_{\text{sat}}/4D$. $\alpha$ is the water vapor

deposition coefficient on ice, $v_{\text{th}}$ is the thermal speed, $n_{\text{sat}}$ is the water vapor number density at ice saturation, and $D$ is the water

vapor diffusion coefficient from the gas phase to the ice phase (Kärcher et al., 2006).





Under a given $R_{seed}$, $W_{seed}$ increases with increasing $N_{seed}$. That is, the more ICs that are added, the more they would reduce $W$. The minimal $N_{seed}$ (i.e., $N_{seedopt}$) is calculated based on the minimal $W_{seed}$, which can prevent homogeneous ice nucleation from

occurring. The default ice nucleation parameterization (Liu and Penner, 2005; LP parameterization) provides a threshold updraft velocity ($W_{thre}$) for homogeneous ice nucleation,

$$W_{thre} = e^{\frac{T-b}{a}}, \tag{A7}$$

where $T$ is the ambient temperature, $a = -1.4938\ln N_{INP} + 12.884$, $b = -10.41\ln N_{INP} - 67.69$. $N_{INP}$ is the INP (e.g., dust aerosol particle) number concentration. Homogeneous ice nucleation does not occur (i.e., only heterogeneous nucleation) if the

efficient updraft velocity ($W_{eff}$, $W_{eff} = W - W_{pre} - W_{seed}$) is less than $W_{thre}$. Thus, the minimal $W_{seed}$ is calculated as $W_{seed} = W - W_{pre} - W_{thre}$. If $W_{seed} < 0$, there is no need for seeding. The minimal number concentration of seeding ICs (i.e., $N_{seedopt}$) can be calculated based on Eq. (A5) and (A6) at the threshold $S_i$ for homogeneous freezing ($S_{ihom}$). In this study, with the given $R_{seed}$, $N_{seedopt}$ is given by:

$$N_{seedopt} = \frac{a_1 S_{ihom} m_w}{(a_2 + a_3 S_{ihom}) 4\pi \rho_i} \frac{1 + b_2 R_{seed}}{b_1 R_{seed}^2} (W - W_{pre} - W_{thre}). \tag{A8}$$

Because the impact of deposition growth on pre-existing ICs is neglected in calculating $W_{pre}$ (Barahona et al., 2014; Shi et al., 2015), the increase in $R_{seed}$ caused by deposition growth during the ice nucleation process is also neglected. As a result, $N_{seedopt}$ might be overestimated, especially for a small given $R_{seed}$. The LP parameterization provides a critical number concentration of INPs ($N_{lim}$) for the only heterogeneous freezing scenario. The $N_{seedopt}$ cannot exceed the $N_{lim}$ because ICs are superior to INPs for hindering homogenous nucleation.

**Code and data availability**

The modified code of CAM5 and the output data used in this study are available online at DOI: 10.5281/zenodo.4507001.

**Competing interests**

The authors declare that they have no conflict of interest.

**Author contribution**

Jiaojiao derived the formula of optimal seeding number concentrations. Xiangjun and Jiaojiao designed the model experiments and developed the model code. Jiaojiao processed and analyzed the raw model output data and wrote the paper. Xiangjun helped to explain the results. Both authors contributed to improving and reviewing the manuscript.





## Acknowledgments

The authors would like to thank Blaž Gasparini for suggesting plotting the cirrus seeding effectiveness. The model simulation
was conducted in the High Performance Computing Center of Nanjing University of Information Science & Technology. This
study was supported by the National Key Research and Development Program of China (grant nos. 2018YFC1507001) and
the National Natural Science Foundation of China (grant nos. 41775095 & 42075145).

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
