# Peer review of "Estimating the potential cooling effect of cirrus thinning achieved via the seeding approach"

_Atmospheric Chemistry and Physics, 2021_

## Referee Comment (RC1)

**Review of Liu and Shi, 2021: Estimating the potential cooling effect of cirrus thinning achieved via the seeding approach**

The reviewed manuscript presents climatic impacts of various seeding strategies in CAM5 general circulation model. In particular, the authors focus on a newly developed optimal seeding method that injects just the right amount of ice nucleating particles (or ice crystals, as suggested in the text) to prevent the formation of homogeneous freezing. Interestingly, seeding was found to modify liquid and mixed-phase clouds in ways that counteract part of the climatic cooling effect at cirrus levels. Such adverse effects can be limited by seeding only areas with solar zenith angles larger than 12°.

The manuscript is nicely structured, has a clear message, and presents the model results in a convincing way. The optimal seeding method is an innovative new way of implementing cirrus cloud seeding in ice nucleation schemes. I recommend publication after minor revision.

**General comments:**

I.) Why is homogeneous freezing switched off at temperatures colder than -68°C? I don't think there is enough evidence from observations for such a drastic modelling choice. Does this decrease the uppermost tropospheric ice crystal number and in such way brings the model closer to the observations? Does this condition influence upper tropospheric  $RH_{ice}$ ?

Would cirrus clouds have a stronger warming effect if you allowed freezing also under the coldest temperatures? Would that increase the radiative impact of cirrus seeding?

II.) What is the spatial and temporal variability of seeded ice crystals in the OPT scenario within a certain region/time? E.g. does the optimal seeded ice crystal number remain constant on the 0.5/1/2/7/24/90-day timescale within a certain area? Would the OPT seeding strategy be feasible in real world or in parts of the world, or should we think of it as a purely academic experiment?

Also, please specify whether you seed at every model timestep in INP20 and INP200 strategies. Would the OPT scenario with a decreased seeding frequency still deliver a significant cooling effect?

**Minor comments:**

page 1, line 20: conserving energy => probably meant "saving energy"?

page 1, line 23: ...as a back-up tool to against...

page 2, line 30:

The sentence

"...which allows more longwave radiation to escape into space so that cool the *Earth*..." should be rewritten.

Maybe: which allows more longwave radiation to escape to space, leading to a cooling effect on the planet.

page 2, line 39:

"...and even the strongest cooling effect may not be ideal" What do you mean with this sentence, please explain!

page 2, lines 60-62:

The sentence on lines 60-62 sounds a bit weird and may need to be rewritten. A suggestion: Therefore, seeding the clouds with a few INPs can prevent the Sice to reach the threshold needed for....

page 3, line 63-64:

Not sure if we can draw a direct connection between decreased ice crystal number concentrations and the name "cirrus thinning". I believe it would be more intuitive for the reader to add the intermediate step of decreasing the cirrus cloud optical depth (as a result of less Ni), which is probably where the name thinning is coming from.

page 3, line 93: Could you add a reference for the sentence starting with "*Notably, in terms*..."

Fig.1: Why does the dashed line representing Si for HET simulation stop at about 60%? Shouldn't it continue increasing even after that point?

page 5, lines 136-137:

I am confused by the two sentences here. The first one ("In the INP20 and INP200...") suggests you implemented seeding by simply increasing the  $N_{dust}$  by 20/200 particles L-1. The sentence "Note that  $N_{dust...}$ " suggests exactly the opposite. So that you keep the  $N_{dust}$  unchanged and you modify the  $N_{seed}$ . Could you therefore explain more thoroughly how did you introduce the seeding particles?

page 8, lines 97-98: Does this mean that in CAM model a substantial fraction of mixed-phase clouds is formed by sedimenting cirrus?

page 8, line 200-201: Why - Are the regions where IWP increases connected to a more vigorous deep convective activity?

page 10, line 239:

"...is used to show how many proportions of iCRE are eliminated..." please rewrite, for instance as: ...is used to show what proportion of iCRE is eliminated...

page 12, title 3.2: It's probably better to use "with" instead of "regarding". page 12:

Another motivation for R10 experiment is probably the smaller mass of the 10 micron ice crystals, compared with the 50 micron one (?)

page 15, line 350: regarding the => with the/using the

page 15, line 353: the global cool effect => the global cooling effect

page 15, line 354-355:

The sentence starting with "*Because there are*…" should be rewritten. Maybe: Avoiding seeding over low-latitude regions can limit some warming effects due to changes in mixed-phase and liquid clouds and thus lead to a more pronounced global cooling effect.

page 16, line 360: maybe better: The overall cooling effect was maximized when using a solar zenith angle threshold of 12°.

page 16, line 363: the sentence "*It is still possible*…" should be reworded. Maybe: The global cooling effect can thus be maximized when limiting seeding to most suitable regions and times of the year.

page 16, line 364: "...in the next step" => in our future work.

page 17, line 395: efficient => effective (?)

---

## Author Comment (AC1)

**Reviewer 1**

We thank the reviewer for the time spent evaluating our study and for the valuable comments and suggestions, which helped us to substantially improve the manuscript. To address the reviewer's comments, some supplemental experiments were carried out. These supplemental experiments were run for 3 model years, and the last 2 years were used in the analyses. We hope that the revised manuscript and our response to the comments are satisfactory. The reviewer's comments are in italics, and our responses are in standard font below.

***General Assessment:***

*The reviewed manuscript presents climatic impacts of various seeding strategies in CAM5 general circulation model. In particular, the authors focus on a newly developed optimal seeding method that injects just the right amount of ice nucleating particles (or ice crystals, as suggested in the text) to prevent the formation of homogeneous freezing. Interestingly, seeding was found to modify liquid and mixedphase clouds in ways that counteract part of the climatic cooling effect at cirrus levels. Such adverse effects can be limited by seeding only areas with solar zenith angles larger than 12°.*

*The manuscript is nicely structured, has a clear message, and presents the model results in a convincing way. The optimal seeding method is an innovative new way of implementing cirrus cloud seeding in ice nucleation schemes. I recommend publication after minor revision.*

Reply: We do appreciate the positive comment.

***General comments:***

*1.) Why is homogeneous freezing switched off at temperatures colder than -68°C? I don't think there is enough evidence from observations for such a drastic modelling choice. Does this decrease the uppermost tropospheric ice crystal number and in such way brings the model closer to the observations? Does this condition influence upper tropospheric RHice?*

*Would cirrus clouds have a stronger warming effect if you allowed freezing also under the coldest temperatures? Would that increase the radiative impact of cirrus seeding?*

Reply: At temperatures below 205 K, the observed ice number concentration ($N_i$) is usually in the range of 10-80 $L^{-1}$ (Krämer et al., 2009), whereas the modeled $N_i$ without switching off homogeneous freezing is usually in the range of 50-2000 $L^{-1}$ (Shi et al., 2015). Because the theory for homogeneous freezing of aqueous aerosols predicts relatively high $N_i$ for cold environments, the $N_i$ from homogeneous freezing should be limited (Jensen et al., 2010). Some recent studies have shown that aerosols rich in organic matter may become glassy (i.e., potential ice nuclei particles) below the glassy-transition temperature (~205 K), and then prevent homogeneous nucleation (Murray et al., 2010; Shiraiwa et al., 2017). Furthermore, a previous study showed that the modeled $N_i$ would be close to the in-situ observation if homogeneous freezing at temperatures colder than the glassy-transition temperature were artificially switched off (Fig. 6 of Shi et al., 2013). In this study, the main purpose of this artificial setting is to make the modeled $N_i$ to be close to observations at temperatures below 205 K. We mentioned this purpose in the revised manuscript.

If homogeneous nucleation is allowed under 205 K, $N_i$ increases over there. Correspondingly, cirrus clouds have a stronger warming effect. However, the cooling effect caused by cirrus seeding is decreased (i.e., weaker cooling effect). Figure S1 shows the comparison between the experiments with (REF and OPT experiments) and without the limitation (REFnolimit and OPTnolimit experiments) of homogeneous freezing under 205 K (hereafter limitation). As expected, the ice cloud radiative effect (iCRE) from the REFnolimit experiment (6.68 W m$^{-2}$) is slightly increased (i.e., stronger warming effect) compared to that from the REF experiment (6.48 W m$^{-2}$) because $N_i$ is increased under 205 K (not shown). The cooling effect from ice clouds ($\Delta$iCRE) caused by cirrus seeding without the limitation (OPTnolimit − REFnolimit, −3.16 W m$^{-2}$) is also slightly stronger than that with the limitation (OPT − REF, −3.04 W m$^{-2}$). The cirrus seeding effectiveness from the OPTnolimit experiment (49.91%) is close to that from the OPT experiment (49.33%). However, the global cooling effect ($\Delta$CRE, from all clouds) from the OPTnolimit experiment (OPTnolimit − REFnolimit, −1.11 W m$^{-2}$) is obviously weaker than that from the OPT experiment (OPT − REF, −1.35 W m$^{-2}$). As introduced in the manuscript, cloud seeding leads to a significant warming effect of liquid and mixed-phase clouds, which counteracts the cooling effect of cirrus clouds. This counteraction becomes stronger in the OPTnolimit experiment.

[Figure]

Figure S1: The annual mean spatial distribution of the ice cloud radiative effect (iCRE, second row) and all cloud radiative effect (CRE, third row) from the REF and REFnolimit experiments (left panel) and the differences from the OPT and OPTnolimit experiments (right panel) with respect to the REF and REFnolimit experiments. The cirrus seeding effectiveness are shown in the first row. Global mean values are shown in the upper right corner. Notably, only 2-year simulation results from each experiment are analyzed here.

*2.) What is the spatial and temporal variability of seeded ice crystals in the OPT scenario within a certain region/time? E.g. does the optimal seeded ice crystal number remain constant on the 0.5/1/2/7/24/90-day timescale within a certain area? Would the OPT seeding strategy be feasible in real world or in parts of the world, or should we think of it as a purely academic experiment?*

*Also, please specify whether you seed at every model timestep in INP20 and INP200 strategies. Would the OPT scenario with a decreased seeding frequency still deliver a significant cooling effect?*

Reply: Thank you for the comments. Figure S2 shows the number concentration of seeding ice crystals

($N_\text{seedopt}$) on the 0.5-, 2-, and 90-day timescales over three different model grids. The $N_\text{seedopt}$ over different regions/times are different. Notably, $N_\text{seedopt}$ is a function of the ambient atmospheric condition and the given radius of seeding ice crystals ($R_\text{seed}$). Therefore, $N_\text{seedopt}$ can be calculated from the current atmospheric conditions or roughly calculated based on the numerical weather forecast for the next few hours. How to determine $N_\text{seedopt}$ may not be the main barrier to the feasibility of the OPT seeding strategy in the real world. However, many aircrafts are needed to add ice crystals at specified times and locations, which is the main barrier to the OPT seeding strategy. This study is a pure academic experiment. The main purpose of this study is to estimate the potential cooling effect of cirrus thinning.

[Figure]

Figure S2: The vertical distribution of $N_\text{seedopt}$ over three model grids, 120 °E and 45 °N (leftmost column), 160 °E and 30 °S (middle column), and 180 °E and 4 °N (rightmost column) on the 0.5- (upper), 2- (middle), and 90-day (lower) timescales. The horizontal coordinate axis is the model time (unit: day).

In the INP20 and INP200 experiments, ice nuclei particles (INPs) are seeded at every model time step, but the seeding INPs only impact the ice nucleation process. This seeding strategy was emphasized in the experimental setups in the revised manuscript.

Similar to the OPT experiment, one more experiment (Reduce experiment) that randomly canceled the seeding operation was carried out. The seeding frequency from the Reduce experiment is approximately half of that from the OPT experiment (Fig. S3). In the Reduce experiment, $\Delta iCRE$ and $\Delta CRE$ are $-1.59$ and $-0.91$ W m$^{-2}$, respectively (Fig. S4). The global cooling effects from the Reduce experiment are weaker than those from the OPT experiment (the 2-year averaged $\Delta iCRE$ and $\Delta CRE$ are $-3.04$ and $-1.35$ W m$^{-2}$, respectively, Fig. S4).

[Figure]

Figure S3: Annual zonal mean and 233 hPa spatial distribution of seeding frequency ($F_{seed}$). The names of the experiments are shown in the upper left corner, and global mean values are shown in the upper right corner. Notably, only 2-year simulation results from each experiment are analyzed here.

[Figure]

Figure S4: The annual mean spatial distribution of differences in iCRE and CRE from the OPT and Reduce experiments with respect to the REF experiment. The cirrus seeding effectiveness are shown in the leftmost column. Global mean values are shown in the upper right corner. Notably, only 2-year simulation results from each experiment are analyzed here.

***Specific comments:***

*3.) page 1, line 20: conserving energy => probably meant "saving energy"?*

Reply: Thanks for the comment. Done.

*4.) page 1, line 23: ...as a back-up tool  against...*

Reply: Thanks. Done.

*5.) page 2, line 30: The sentence "...which allows more longwave radiation to escape into space so that cool the Earth..." should be rewritten.*

*Maybe: which allows more longwave radiation to escape to space, leading to a cooling effect on the planet.*

Reply: Thanks. In the revised manuscript, the sentence was rewritten based on this comment.

*6.) page 2, line 39: "...and even the strongest cooling effect may not be ideal" What do you mean with this sentence, please explain!*

Reply: Thank you for this comment. Previous studies have shown that the global cooling effects with fixed seeding number concentrations are not strong enough even for their strongest cooling effects (usually above $-1.0$ W m$^{-2}$). In the revised manuscript, the sentence was rewritten.

*7.) page 2, lines 60-62: The sentence on lines 60-62 sounds a bit weird and may need to be rewritten. A suggestion: Therefore, seeding the clouds with a few INPs can prevent the Sice to reach the threshold needed for....*

Reply: Thanks. In the revised manuscript, the sentence was changed as suggested: "Therefore, seeding the clouds with a few INPs (usually less than 100 L$^{-1}$) can prevent $S_i$ from reaching $S_{ihom}$".

*8.) page 3, line 63-64: Not sure if we can draw a direct connection between decreased ice crystal number concentrations and the name "cirrus thinning". I believe it would be more intuitive for the reader to add the intermediate step of decreasing the cirrus cloud optical depth (as a result of less Ni), which is probably where the name thinning is coming from.*

Reply: Thank you for this helpful comment. To improve the readability, the sentence was rewritten in the revised manuscript: "As a result, the in-cloud IC number concentrations ($N_i$) are usually decreased, which decrease the cirrus cloud optical depth (i.e., cirrus thinning)".

*9.) page 3, line 93: Could you add a reference for the sentence starting with "Notably, in terms..."*

Reply: Thanks. Some references (Pruppacher and Klett, 1998; Kärcher et al., 2006; Shi et al., 2015) were added in the revised manuscript.

*10.) Fig.1: Why does the dashed line representing Si for HET simulation stop at about 60%? Shouldn't it continue increasing even after that point?*

Reply: Yes, the $S_i$ for HET simulation continues to increase throughout the process. In the revised manuscript, the dashed line representing $S_i$ for the HET simulation stops at 80% (i.e., the upper limit of $S_i$ in the figure). The updated figure is as follows:

[Figure]

*11.) page 5, lines 136-137: I am confused by the two sentences here. The first one ("In the INP20 and INP200…") suggests you implemented seeding by simply increasing the N dust by 20/200 particles L-1. The sentence "Note that Ndust…" suggests exactly the opposite. So that you keep the N dust unchanged and you modify the Nseed. Could you therefore explain more thoroughly how did you introduce the seeding particles?*

Reply: Thank you for this comment. We apologize for the confusion. Note that, "$N_{dust}$ used for driving ice nucleation parameterization" and "$N_{dust}$ in the aerosol module" could be two different variables. In the default model, the "$N_{dust}$ used for driving ice nucleation parameterization" is same as the "$N_{dust}$ in the aerosol module". In the INP20/200 experiments of this study, the "$N_{dust}$ in the aerosol module" is not changed, but the "$N_{dust}$ used for driving ice nucleation parameterization" = the "$N_{dust}$ in the aerosol module" + 20/200 particles $L^{-1}$. To avoid confusion, the two sentences were rewritten in the revised manuscript.

*12.) page 8, lines 97-98: Does this mean that in CAM model a substantial fraction of mixed-phase clouds is formed by sedimenting cirrus?*

Reply: Thank you for this comment. Under the condition of 300 hPa and −35 °C, the sedimentation velocity of the 10 μm ice crystal is ~150 m hour$^{-1}$ (~ 4 km day$^{-1}$), and the sedimentation velocity of the 50 μm ice crystal is ~700 m hour$^{-1}$ (~ 18 km day$^{-1}$). The sedimentation process certainly influences the evolution of cirrus clouds. As shown in Fig. S5, the ice mass mixing ratio tendency caused by

sedimentation (QISEDTEN) is negative at cirrus layers but positive at mixed-phase layers. Note that, ice crystals would sublimate if they fell into clear-sky areas. Therefore, Fig. S5 cannot indicate that a substantial fraction of mixed-phase clouds is formed by sedimenting cirrus. To avoid the misleading implication that sedimentation is the main mechanism for mixed-phase clouds formation, analyses of the possible reason for the increase of IWC at mixed-phase layers were added in the revised manuscript as follows: "In the HET and OPT experiments, both $N_i$ and IWC in middle and lower mixed-phase clouds are obviously increased. The main reason might be that the deep convective activity becomes more vigorous because cirrus thinning reduces atmospheric stability via the radiative budget. The ratio of ice to total cloud condensate detrained from the convective parameterizations is a linear function of temperature between −40 °C and −10 °C (Morrison and Gettelman, 2008). Furthermore, the ICs can grow through the Bergeron process in mixed-phase clouds (Morrison and Gettelman, 2008). This might be the reason that the relative increases in IWC in mixed-phase clouds are stronger than the relative increases in $N_i$".

[Figure]

Figure S5: Annual zonal mean distribution of the ice mass mixing ratio tendency caused by sedimentation (QISEDTEN) from the REF experiment.

*13.) page 8, line 200-201: Why - Are the regions where IWP increases connected to a more vigorous deep convective activity?*

Reply: Yes, the regions where IWP increases are connected to more vigorous deep convective activity. Notably, the cloud condensate detrained from the convective parameterizations is considered as ice below −40 °C. More importantly, over these regions, the decreases of IWP induced by cirrus seeding are weak.

*14.) page 10, line 239: "…is used to show how many proportions of iCRE are eliminated…" please rewrite, for instance as: …is used to show what proportion of iCRE is eliminated…*

Reply: Thank you for this comment. In the revised manuscript, the sentence was changed as suggested.

*15.) page 12, title 3.2: It's probably better to use "with" instead of "regarding".*

Reply: Thanks. Done. Furthermore, we checked this expression throughout the manuscript.

*16.) page 12: Another motivation for R10 experiment is probably the smaller mass of the 10 micron ice crystals, compared with the 50 micron one (?)*

Reply: The total mass of seeding ice crystals from the R10 experiment (seeding ice crystal is 10 μm) is significantly smaller than that from the OPT experiment (seeding ice crystal is 50 μm), even if the number concentration of seeding ice crystals from the R10 experiment is twice as high as that from the OPT experiment. In this study, it is assumed that the seeding ice crystals can be made from ambient atmospheric water vapor. Thus, the transportation problem is not considered.

*17.) page 15, line 350: regarding the => with the/using the*

Reply: Thanks. Done.

*18.) page 15, line 353: the global cool effect => the global cooling effect*

Reply: Thanks. Done. We also carefully checked the entire manuscript for grammatical and formatting errors.

*19.) page 15, line 354-355: The sentence starting with "Because there are…" should be rewritten. Maybe: Avoiding seeding over low-latitude regions can limit some warming effects due to changes in mixed-phase and liquid clouds and thus lead to a more pronounced global cooling effect.*

Reply: Thanks. In the revised manuscript, the sentence was changed as suggested.

*20.) page 16, line 360: maybe better: The overall cooling effect was maximized when using a solar zenith angle threshold of 12°.*

Reply: Thanks. In the revised manuscript, the sentence was changed as suggested.

*21.) page 16, line 363: the sentence "It is still possible…" should be reworded. Maybe: The global cooling effect can thus be maximized when limiting seeding to most suitable regions and times of the year.*

Reply: Thanks. In the revised manuscript, the sentence was reworded as suggested.

*22.) page 16, line 364: "…in the next step" => in our future work.*

Reply: Thanks. Done.

*23.) page 17, line 395: efficient => effective (?)*

Reply: Thanks. Done.

**References:**

Jensen, E., Pfister L., Bui T., Lawson P., and Baumgardner D.: Ice nucleation and cloud microphysical properties in tropical tropopause layer cirrus. Atmos. Chem. Phys., 10 (3), 1369–1384, https://doi.org/10.5194/acp-10-1369-2010, 2010.

Kärcher, B., Hendricks, J. and Lohmann, U.: Physically based parameterization of cirrus cloud formation for use in global atmospheric models, J. Geophys. Res., 111(D1), D01205, https://doi.org/10.1029/2005JD006219, 2006.

Krämer, M., Schiller, C., Afchine, A., Bauer, R., Gensch, I., Mangold, A., Schlicht, S., Spelten, N., Sitnikov, N., Borrmann, S., de Reus, M., and Spichtinger, P.: Ice supersaturations and cirrus cloud crystal numbers, Atmos. Chem. Phys., 9, 3505–3522, doi:10.5194/acp-9-3505-2009, 2009.

Morrison, H. and Gettelman, A.: A New Two-Moment Bulk Stratiform Cloud Microphysics Scheme in the Community Atmosphere Model, Version 3 (CAM3). Part I: Description and Numerical Tests, J. Clim., 21(15), 3642–3659, https://doi.org/10.1175/2008JCLI2105.1, 2008.

Murray, B., Wilson, T., Dobbie, S., Cui, Z., Al-Jumur, S., Möhler, O., Schnaiter, M., Wagner, R., Benz, S., Niemand, M., Saathoff, H., Ebert, V., Wagner, S., Kärcher, B.: Heterogeneous nucleation of ice particles on glassy aerosols under cirrus conditions. Nature Geoscience., 3.233-237, 10.1038/ngeo817, 2010.

Pruppacher, H. and Klett, J.: Microphysics of Clouds and Precipitation: second revised and enlarged edition with an introduction to cloud chemistry and cloud electricity, Kluwer, Dordrecht., 1998.

Shi, X., Liu, X. and Zhang, K.: Effects of pre-existing ice crystals on cirrus clouds and comparison between different ice nucleation parameterizations with the Community Atmosphere Model (CAM5), Atmos. Chem. Phys., 15(3), 1503–1520, https://doi.org/10.5194/acp-15-1503-2015, 2015.

Shi, X., Wang, B., Liu, X., and Wang, M.: Two-moment bulk stratiform cloud microphysics in the Grid-point Atmospheric Model of IAP LASG (GAMIL). Adv. Atmos. Sci., 30(3),868–883, doi: 10.1007/s00376-012-2072-1, 2013.

Shiraiwa, M., Li, Y., Tsimpidi, A., Karydis V., Berkemeier T., Pandis S., Lelieveld J., Koop T., and Poeschl U.: Global distribution of particle phase state in atmospheric secondary organic aerosols. Nat Commun., 8, 15002, https://doi.org/10.1038/ncomms15002, 2017.

---

## Author Comment (AC2)

**Reviewer 2**

We appreciate the thorough review and constructive comments, which have helped us to substantially improve the quality of the manuscript. We hope that the modified manuscript and our responses to the comments are satisfactory. The reviewer's comments are in italics, and our responses are in standard font below.

1.) This manuscript documents experiments with variable ice nucleation number, by adjusting the ice concentration in a GCM to prevent homogenous freezing and thin cirrus clouds as a form of geo engineering. The manuscript is well written and thought out. It should be publishable in ACP with minor revisions.

My major concern is a lot of the effects are from liquid and mixed phase clouds that the method is not directly perturbing. Or should not be directly perturbing. But the mechanism is not explained well in the text. It is not clear if the compensating effects for mixed and liquid clouds from adding ice crystals to pure ice clouds is an artifact of the model, or has a solid physical basis. This should be detailed further in the discussion of the cases and the conclusions. It may require a bit more in depth analysis.

**Reply:** Thank you very much for your kind statement. We totally understand the reviewer's concern about the compensating effect (i.e., warming effect of liquid and mixed-phase clouds) caused by cirrus seeding. Gasparini et al. (2017) pointed out the compensating effects induced by cirrus thinning. There seems to be a relatively solid physical reason that cirrus thinning reduces atmospheric stability via the radiative budget and convective activity becomes more intense (Kristjánsson et al., 2015; Storelvmo and Herger, 2014), which would consume more cloud water and lead to the warming effect of liquid clouds (Gasparini et al., 2017; Rapp et al., 2011). The warming effect of liquid clouds in our study ( $0.94 \pm 0.11$  W m-2 from the OPT experiment) is similar to that reported by Gasparini et al. (2017; Table 5,  $0.96 \pm 0.25$  W m-2 from the ECHAM-HAM model simulation with seeding 1 ice nuclei particle L-1 of 50 µm). However, the warming effect of mixed-phase clouds in our study ( $1.09 \pm 0.11$  W m-2) is several times stronger than that from their results (Table 5,  $0.15 \pm 0.10$  W m-2). In our study (using the CAM model), the warming effect of mixed-phase clouds mainly comes from the increasing longwave component (not shown), which is consistent with the increased ice water content (IWC) in mixed-phase clouds. The main reason for the

increased IWC might be that the convective detrainment is obviously increased due to cirrus thinning. The ratio of ice to total cloud condensate detrained from the convective parameterizations is a linear function of temperature between -40 °C and -10 °C (Morrison and Gettelman, 2008). Furthermore, ice crystals can grow through the Bergeron process in mixed-phase clouds (Morrison and Gettelman, 2008). The warming effect of mixed-phase clouds induced by cirrus thinning seems more sensitive to the cloud-related scheme used in the climate model. In the reversed manuscript, we added more analyses about the compensating effect (i.e., the warming effect of mixed-phase and liquid clouds) in Section 3. In the discussion section, we clearly pointed out that the climatic response to cirrus seeding is complex and might differ among different climate models and seeding methods. The compensating effects introduced in this study are derived from the atmosphere-only simulations with prescribed ocean surface conditions, the coupled model simulations might show different results (e.g., Gasparini et al., 2017).

**Specific comments:**

2.) Page 1, L11: is cooling positive or negative? Do the values in lines 10 and 11 with different signs represent the same direction? Please clarify: if the same direction then should be the same sign.

**Reply**: We thank the reviewer for pointing this out. Cooling should be negative even if it has been declared a cooling effect. A negative sign was added in the revised manuscript. Furthermore, we have checked the signs throughout the manuscript and emphasized this issue in the revised manuscript.

**3.) Page 1, L14: cirrus clouds typically do not contain liquid, so why would cirrus seeding impact mixed phase clouds?**

**Reply**: Thank you for this comment. Cirrus seeding can directly reduce the cirrus clouds and then indirectly impact mixed-phase and liquid clouds via different mechanisms. For instance, convective activity becomes more intense because cirrus thinning reduces atmospheric stability via the radiative budget, and the increased convective activity could impact liquid and mixed-phase clouds.

**4.) Page 1, L16: could yield. Also, what sign is intended here?**

Reply: Thanks. Done. The cooling effect is always quantified by a negative value in the revised manuscript.

**5.) Page 3, L80: this seems like a pretty significant change to shut off homogeneous freezing below 205K. What is the impact of that?**

**Reply:** Thank you for this comment. At temperatures below 205 K, the observed ice number concentration  $(N_i)$  is usually in the range of 10-80 L-1 (Krämer et al., 2009), whereas the modeled  $N_i$  without switching off homogeneous freezing is usually in the range of 50-2000 L-1 (Shi et al., 2015). In this study, the main purpose of this artificial setting is to make the modeled  $N_i$  to be close to observations at temperatures below 205 K (Shi et al., 2013). We mentioned this purpose in the revised manuscript.

**6.) Page 4, L97: I think the formulas are an important part of the study, and should be in the main text.**

**Reply:** Thanks for this suggestion. The flexible seeding method used for estimating the potential cooling effect of cirrus thinning includes the optimal seeding number concentration and the flexible seeding strategy. In the main text, we focus on introducing the advantages of the flexible seeding method, which are important for estimating the potential cooling effect of cirrus thinning. The formulas for calculating the optimal seeding number concentration provided by this study depend on the ice nucleation parameterization used in the climate model. However, the design idea of the optimal seeding number concentration  $(N_{\rm lim})$  of ice nuclei particles (INPs) for the only heterogeneous freezing scenario (e.g., Barahona and Nenes, 2009).  $N_{\rm lim}$  could also be used as the optimal seeding number concentration with INPs. Therefore, we prefer to focus on the design idea of the flexible seeding number concentration with INPs.

7.) Page 5, L134: if you add ice number do you have to worry about conservation or inconsistencies? Not for number, but do such affects bleed into humans.

**Reply:** In the cirrus seeding experiments in this study, the seeding ice crystals are made from ambient atmospheric water vapor rather than artificially adding water into the atmosphere. Therefore, it is unnecessary to consider conservation or inconsistencies. We emphasized this in the experimental setups of the revised manuscript.

**8.) Page 5, L140: does the ice added feedback through the microphysics?**

**Reply**: Yes, it does. The seeding ice crystals not only impact the ice nucleation process but also impact other cloud microphysics processes because the ice crystals are directly added to the microphysics scheme. As shown in Section 3 of the manuscript, the impacts of cirrus seeding are very complicated. There is not only the direct instantaneous impact but also the feedback through microphysics.

9.) Page 10, L226: for clarity maybe you could pull out the global mean values into a table or a set of histograms.

Reply: Thank you for the comment. A new table was added in the revised manuscript as follows:

Table 2. Global annual mean cloud radiative effects from all experiments a. The corresponding standard deviations calculated from the difference of each year for 10 years are shown in brackets.

| Experiments                      | REF    | HET-REF     | OPT-REF     | INP20-REF   | INP200-REF  | R10-REF     | GT-REF      |
|----------------------------------|--------|-------------|-------------|-------------|-------------|-------------|-------------|
| iCREsw (W m -2 )      | -5.30  | 3.39(0.03)  | 3.25(0.05)  | 0.38(0.07)  | 0.30(0.05)  | 2.81(0.05)  | 1.99(0.04)  |
| $iCRE_{LW}$ (W m -2 ) | 11.79  | -6.84(0.04) | -6.29(0.07) | -0.83(0.10) | -0.31(0.08) | -5.40(0.07) | -4.33(0.06) |
| iCRE (W m -2 )        | 6.49   | -3.45(0.02) | -3.04(0.03) | -0.44(0.04) | -0.01(0.04) | -2.58(0.03) | -2.34(0.03) |
| Effectiveness (%)                |        | 56.19(0.70) | 49.40(0.62) | 6.69(1.45)  | -2.22(1.32) | 43.02(0.85) | 39.01(0.95) |
| mCRE (W $m^{-2}$ )               | -6.20  | 1.06(0.13)  | 1.09(0.11)  | 0.20(0.11)  | 0.15(0.13)  | 0.90(0.10)  | 0.81(0.12)  |
| ICRE (W m -2 )        | -24.69 | 1.06(0.14)  | 0.94(0.11)  | 0.07(0.17)  | -0.07(0.13) | 0.62(0.13)  | 0.03(0.17)  |
| CRE (W m -2 )         | -28.43 | -1.98(0.26) | -1.36(0.18) | -0.27(0.26) | 0.35(0.28)  | -1.25(0.22) | -2.00(0.25) |

a Shown are the ice cloud shortwave radiative effect (iCREsw), ice cloud longwave radiative effect (iCRELW), ice cloud radiative effect (iCRE), cirrus seeding effectiveness (Effectiveness), mixed-phase cloud radiative effect (mCRE), liquid cloud radiative effect (ICRE), and all cloud radiative effect (CRE).

10.) Page 10, L240: please name and make a formal equation for the seeding effectiveness.

**Reply**: Thanks. The cirrus seeding effectiveness ( $-100 * |\Delta iCRE / iCRE|$ ) was first proposed in Gasparini et al. (2020), which was used to show what proportion of iCRE is eliminated by cirrus seeding. In the revised manuscript, the equation has been clarified.

11.) Page 12, L274: can you comment more on the mechanism: why does adding ICs impact mixed phase and liquid clouds? Is this a real effect or a feature of the model formulation?

**Reply**: As discussed above, the impact on liquid clouds seems to be a real effect, and the impact on mixed phase clouds might be an artifact of the bulk model physics. In the revised manuscript, further comments on the mechanism were given in Section 3 (experimental results), and the robustness of the mechanism was analyzed in the discussion section.

**12.) Page 12, L283: can you explain in a sentence or so why N increases with decreasing R?**

**Reply**: The large ice crystal (R is relatively larger) has a larger surface area, consuming water vapor via deposition growth more efficiently. For the same amount of ice crystals, it is easier for larger ice crystals to prevent homogeneous nucleation from happening.

**13.) Page 12, L284: which experiments are "these"?**

Reply: Thank you for this comment. "In these seeding experiments" refers to the OPT, R10 and GT experiments. In the revised manuscript, we have clarified this.

**14.) Page 12, L285-7: Im lost here. I thought the point of seeding was not to reach Sihom? Please clarify.**

Reply: Yes, the point of seeding was not to reach  $S_{\text{ihom}}$ . Notably, seeding occurs only where homogeneous nucleation would occur without seeding. This seeding strategy was clearly described in Section 2.3 (flexible seeding method).

**15.) Page 14, L314: can you explain why? Does it have to do with liquid and mixed phase clouds?**

**Reply**: As shown in Fig. S1, the cooling effect of cirrus seeding is more significant in the winter hemisphere. In low solar noon zenith angle regions, the warming effects from liquid clouds are pronounced because the water vapor is sufficient and convective activity is intense. Thus, avoiding seeding in low solar noon zenith angle regions (GT experiment in the manuscript) can obtain a stronger cooling effect.

Figure S1: The annual mean spatial distribution of all cloud radiative effects (CREs) from the REF experiment for DJF and JJA (left panel) and the differences between the OPT and REF experiments (right panel). Global mean values are shown in the upper right corner, and the corresponding standard deviations calculated from the difference of each year for 10 years are shown in brackets.

**16.) Page 14, L324: but don't you need INP to make IC?**

Reply: It can be assumed that there is a machine that can make ice cubes from atmospheric water vapor and break the ice cubes into ice crystals.

17.) Page 15, L311: This is attributed to...

Reply: Thanks. Done.

18.) Page 15, L342: what is the mechanism for the effects on liquid and mixed phase clouds? You need to figure out of this is real or an artifact of the bulk model physics.

**Reply:** There might be many mechanisms for the effects on liquid and mixed-phase clouds. In the revised manuscript, the main possible mechanism was given. However, based on simulation results from one climate model, it is difficult to determine whether this is a real or an artifact of the bulk model physics. As discussed above, the comparison between our study and the study of Gasparini et al. (2017) suggests that the mechanism for the effect on liquid clouds might be a real physical mechanism, and the mechanism for the effects on mixed-phase clouds might be an artifact of the bulk model physics.

19.) Page 15, L353: is sedimentation the mechanism then? Can you test that?

**Reply:** The mechanisms for the warming effects from mixed-phase and liquid clouds are complicated. The increased convective activity might be the main mechanism, as discussed in Gasparini et al. (2017). Compared with the 50 µm ice crystal, the sedimentation of the 10 µm ice crystal is slower (i.e., longer lifetime in cirrus). In addition,  $N_{\text{seedopt}}$  from the R10 experiments (seeding ice crystal is 10 µm) is larger than that from the OPT experiment (seeding ice crystal is 50 µm). As a result, the decrease in the cirrus cloud radiative effect from the R10 experiments is weaker than that from the OPT experiment. Correspondingly, the increase in convective activity from the R10 experiment is weaker than that from the OPT experiment. To avoid misleading that sedimentation is the main mechanism of mixed-phase clouds formation, the sentence has been reworded.

20.) Page 17, L388: you say ICs increase W, but then contradict that in the next phase. This is confusing and unclear.

Reply: Thank you for this comment. Equation (A3) shows that the effect of pre-existing ICs ( $W_{pre}$ ) and seeding ICs ( $W_{\text{seed}}$ ) on ice nucleation can be taken as reducing the vertical velocity used for driving the ice nucleation parameterization. The effective updraft velocity ( $W_{eff}$ ) used for ice nucleation is calculated as  $W_{\rm eff} = W - W_{\rm pre} - W_{\rm seed}$ , where W is the ambient updraft velocity. Increasing seeding ICs would increase the  $W_{\text{seed}}$ , and W would decrease.

**References:**

- Barahona, D. and Nenes, A.: Parameterizing the competition between homogeneous and heterogeneous freezing in ice cloud formation – polydisperse ice nuclei, Atmos. Chem. Phys., 9(16), 5933–5948, https://doi.org/10.5194/acp-9-5933-2009, 2009.
- Gasparini, B., Münch, S., Poncet, L., Feldmann, M. and Lohmann, U.: Is increasing ice crystal sedimentation velocity in geoengineering simulations a good proxy for cirrus cloud seeding?, Atmos. Chem. Phys., 17(7), 4871–4885, https://doi.org/10.5194/acp-17-4871-2017, 2017.
- Gasparini, B., McGraw, Z., Storelvmo, T. and Lohmann, U.: To what extent can cirrus cloud seeding counteract global warming?, Environ. Res. Lett., 15(5), 054002, https://doi.org/10.1088/1748-9326/ab71a3, 2020.
- Krämer, M., Schiller, C., Afchine, A., Bauer, R., Gensch, I., Mangold, A., Schlicht, S., Spelten, N., Sitnikov, N., Borrmann, S., de Reus, M., and Spichtinger, P.: Ice supersaturations and cirrus cloud crystal numbers, Atmos. Chem. Phys., 9, 3505–3522, doi:10.5194/acp-9-3505-2009, 2009. Kristjánsson, J. E., Muri, H. and Schmidt, H.: The hydrological cycle response to cirrus cloud thinning,
- Geophys. Res. Lett., 42(24), 10,807-10,815, https://doi.org/10.1002/2015GL066795, 2015.
- Morrison, H. and Gettelman, A.: A New Two-Moment Bulk Stratiform Cloud Microphysics Scheme in the Community Atmosphere Model, Version 3 (CAM3). Part I: Description and Numerical Tests, J. Clim., 21(15), 3642–3659, https://doi.org/10.1175/2008JCLI2105.1, 2008.
- Rapp, A., Kummerow, C., and Fowler, L.: Interactions between warm rain clouds and atmospheric preconditioning for deep convection in the tropics. J. Geophys. Res. Atmos., 116, D23210, https://doi.org/10.1029/2011JD016143, 2011.

- Shi, X., Wang, B., Liu, X., and Wang, M.: Two-moment bulk stratiform cloud microphysics in the Gridpoint Atmospheric Model of IAP LASG (GAMIL). Adv. Atmos. Sci., 30(3),868–883, doi: 10.1007/s00376-012-2072-1, 2013.
- Shi, X., Liu, X. and Zhang, K.: Effects of pre-existing ice crystals on cirrus clouds and comparison between different ice nucleation parameterizations with the Community Atmosphere Model (CAM5), Atmos. Chem. Phys., 15(3), 1503–1520, https://doi.org/10.5194/acp-15-1503-2015, 2015.
- Storelvmo, T. and Herger, N.: Cirrus cloud susceptibility to the injection of ice nuclei in the upper troposphere, J. Geophys. Res. Atmos., 119(5), 2375–2389, https://doi.org/10.1002/2013JD020816, 2014.